# Design-driven optimization of low-cost reagent formulations for reproducible and high-yielding cell-free gene expression

Meagan L. Olsen [1,2], Caroline E. Copeland[3], Chad A. Sundberg[4,5], Rochelle Aw [6,7], Zachary M. Shaver [2,8,9], Govind Rao [4,5], James R. Swartz [3], Ashty S. Karim [1,2] ✉ & Michael C. Jewett [1,2,6] ✉

Access to recombinant proteins is vital in basic science and biotechnology research. Cell-free gene expression systems provide one approach to address this need, but widespread utilization remains limited by the cost, complexity, and inconsistency of current platforms. To address these limitations, we carry out a multi-dimensional definitive screening design to reduce the number of reagent components and remove costly secondary energy substrates. From 1,231 different reagent formulations, we discover a simple and reproducible system based on 12 components. The optimized reagent formulation can produce $2.4 \pm 0.3$ g/L of protein product at the 15-μL scale (~$60/g_{protein}$) and $3.7 \pm 0.2$ g/L (~$39/g_{protein}$) at the 4-mL scale with oxygen supplementation. This provides an average 95% reduction in cost over previous cell-free reagent formulations. We further show that the optimized reagent formulation can produce nucleoside triphosphates from nitrogenous bases and ribose and that it is robust to failure across batches of cell lysates, users/locations, and in the synthesis of more than 20 different proteins. For example, we demonstrate the production of fifteen therapeutically relevant products, including full-length aglycosylated monoclonal antibodies. We anticipate that our optimized reagent formulation will democratize the use of cell-free systems for protein manufacturing and synthetic biology applications.

Cell-free gene expression (CFE) systems can produce recombinant proteins in cell lysates upon incubation with essential substrates (e.g., amino acids, DNA template, energy substrates)[1–3]. These systems have advanced applications in diagnostics[4–16], pathway prototyping[17–22], protein design[23–27], metabolic engineering[28–32], and education[33–38]. Cell-free systems provide numerous benefits for protein production that are complementary to cell-based production platforms. First, cell-free

systems are modular due to their open reaction environment. New functionalities (e.g., post-translational modifications) can be added through drop-in reagents. In addition, different protein products can be made using the same biocatalyst (i.e., same cell lysate) by changing the encoded DNA added to the reaction; the lysate source strain does not need to be re-engineered. Second, cell-free systems are scalable from the nL to the 1,000-L scale[39] with growth-independent batch-to-

[1]Department of Chemical and Biological Engineering, Northwestern University, Evanston, IL, USA. [2]Center for Synthetic Biology, Northwestern University, Evanston, IL, USA. [3]Department of Chemical Engineering, Stanford University, Stanford, CA, USA. [4]Department of Chemical, Biochemical and Environmental Engineering, University of Maryland, Baltimore County, Baltimore, MD, USA. [5]Center for Advanced Sensor Technology, University of Maryland, Baltimore County, Baltimore, MD, USA. [6]Department of Bioengineering, Stanford University, Stanford, CA, USA. [7]School of Life Sciences, University of Nottingham, Nottingham, UK. [8]Interdisciplinary Biological Sciences Graduate Program, Northwestern University, Evanston, IL, USA. [9]Medical Scientist Training Program, Northwestern University, Chicago, IL, USA. ✉e-mail: ashty.karim@northwestern.edu; mjewett@stanford.edu

batch performance. Third, distributed manufacturing paradigms can be enabled by cell-free systems, which can be freeze-dried, distributed, stored, and then readily reactivated by just adding water[40–43]. For such purposes, state-of-the-art cell-free expression systems can produce a wide variety of biologics, including full-length aglycosylated monoclonal antibodies and glycosylated therapeutics[23,27,40–42,44–56].

Despite the growing use of cell-free systems, reagent costs remain expensive for widespread commercial adoption. Cell-free gene expression, for example, depends on small-molecule reagent components that include secondary energy substrates (e.g., phosphoenolpyruvate); DNA, RNA, and protein building blocks (e.g., NTPs, amino acids); cofactors and coenzymes (e.g., CoA, NAD); salts (e.g., magnesium); polyamines; and other chemical components (e.g., buffers). Many reagent formulations rely on expensive phosphorylated energy substrates (e.g., \$2,000-3,000/$L_{CFE}$), such as phosphoenolpyruvate (PEP), to regenerate the ATP that provides energy for protein biosynthesis[57–59]. This results in reagent costs upwards of \$4,000/$L_{CFE}$ and contributes to roughly 75% of all material costs associated with cell-free expression (Table 1; Supplementary Table 1). While one report suggests cell-free systems can produce upwards of 4 g/L of protein in batch operation[60], most studies report < 2 g/L of protein[1], which leaves costs an order of magnitude higher than the \$10-\$100/$g_{protein}$ associated with cell-based production[61,62]. Manufacturer availability and variability of complex reagents like purified tRNA further complicate implementation[63].

To address high costs of cell-free gene expression, numerous groups have developed different reagent formulations that leverage less expensive non-phosphorylated secondary energy substrates for reagent costs of < \$500/$L_{CFE}$ (Table 1)[41,45,57,64–67]. However, these reagent systems have not been widely adopted. As such, reducing costs while improving protein yields remains essential for commercial use of cell-free protein manufacturing strategies.

In this work, we aimed to develop a cell-free expression reagent formulation that enables low-cost, consistent, and high-yield protein production. A key feature of our approach was the focus on reducing the number of components used while simultaneously designing for yield and cost metrics. The goal was to achieve less than \$100 in reagents per gram of protein to more closely match cell-based production costs[61,62]. Inspired by previous multi-dimensional optimization efforts[68,69], we carried out design-driven optimization to screen 58 cell-free reagent components and their concentrations in 1,231 different reaction combinations. Our most productive formulation was able to synthesize 2.4 ± 0.3 g/L superfolder green fluorescent protein (sfGFP) at a reagent cost of \$143/$L_{CFE}$ (\$60/$g_{protein}$) in a standard batch reaction. Improving oxygen flux further increased yields to 3.7 ± 0.2 g/L sfGFP (\$39/$g_{protein}$). This optimized formulation lowers cell-free expression cost by an average of 95% over state-of-the-art phosphorylated energy substrate formulations. Lastly, we demonstrate modularity for protein manufacturing by using this reagent formulation to produce more than 20 recombinant proteins, including fifteen recombinant protein therapeutics (e.g., full-length aglycosylated monoclonal antibodies and other disulfide bonded products). We anticipate that low-cost, consistent, and high-yield cell-free systems will expand existing biomanufacturing efforts, helping to meet the need for recombinant proteins.

## Results
### Comparing state-of-the-art cell-free reagent formulations
We first set out to create a baseline comparison of previously reported reagent formulations by experimentally examining their protein production capacity in our laboratory (Table 1). By doing this, we can fairly compare the value of reagent differences in these formulations. Specifically, we tested nine previously reported reagent formulations for *E. coli* crude lysate platforms[41,45,57,58,64,65], focusing on low-cost variants that leverage non-phosphorylated energy substrates like glucose or

glutamate. This included the best performing formulations identified by two machine learning-based optimization approaches[68,70]. We used an *E. coli* BL21 Star (DE3) lysate to test each formulation, though some of the formulations were originally designed around other B- and K-strains of *E. coli*. We individually assembled each formulation, constructed 15-μL cell-free gene expression reactions containing pJL1-sfGFP plasmid DNA, and incubated reactions at 30 °C for 20 h. We found that the PANOx-SP formulation (our phosphorylated energy substrate base case)[57] produced the highest sfGFP yield and used this formulation as a comparison standard for the remainder of the work. Although all non-phosphorylated energy substrate systems produced less total protein than the phosphorylated substrate formulations, the reagent cost per gram of protein ranged from 22-92% less than that of the PANOx-SP system (Table 1; and Fig. 1a).

### Optimizing a cell-free reagent formulation
Building on these established reagent formulations, we set out to identify a low-cost, consistent, and high-yielding cell-free reagent formulation (i.e., minimize \$$_{reagent}$/$g_{protein}$) through exploration and optimization (Supplementary Figs. 1, 2; and Supplementary Data Files 1, 2). We began with a panel of 11 reagent components common amongst the established reagent formulations. These components included magnesium glutamate, potassium glutamate, the 20 standard amino acids (counted as a single component), phosphate, oxalate, spermidine, and oxidized glutathione. We varied these components while maintaining cell extract and plasmid DNA at fixed concentrations. We started by identifying upper and lower concentration bounds for each component at which we observed a measurable level of sfGFP synthesis. Using these concentration ranges, we then carried out a Definitive Screening Design (DSD) to define 16 reaction formulations using the 11 components. We tested these formulations in the laboratory and fit a stepwise model with the resulting data. We next adjusted formulation ranges based on components the model found significant. We repeated the process across 7 experiments to test a total of 67 formulations identified by Design of Experiments methods, 5 formulations predicted to be optimal by the stepwise models, and an additional 75 formulations designed based on reagent solubility and observed behaviors (Stage 1a; Figs. S1, S2; and Supplementary Data File 1). After 7 rounds and 147 total formulations (Fig. 1b), we identified an active reagent formulation (selected based on activity and minimized number of components) comprised only of potassium glutamate, nucleoside monophosphates, and amino acids. A three-factor optimization yielded a formulation that produced 0.5 ± 0.1 g/L sfGFP (Fig. 1c; and Supplementary Fig. 3). This minimal system (\$175/$g_{protein}$) is the smallest set of added reagent components reported for any in vitro, combined transcription and translation system.

Next, we explored supplementation of other reagents canonically used in cell-free gene expression to our minimal system to see if we could improve total protein yields while keeping the number of reagents low. We constrained our search to lower cost reagents (i.e., less than \$50/$g_{reagent}$) to balance the need for low-cost formulations with potential increased protein yields. Many of the more expensive reagents, including tRNA, coenzyme A, and NAD$^+$, have previously been shown to be unnecessary for cell-free gene expression when cell lysates are not dialyzed[41,45], so we did not include them here. After setting this initial constraint, all further optimization was evaluated based on formulation productivity except where noted. Inspired by previous multi-dimensional reagent optimization efforts[68–70], we tested new reagent formulations (based on the minimal formulation) in parallel and independent experimental rounds defined by (i) Design of Experiments as described in our initial exploration, (ii) the METIS active learning platform[59], and (iii) our domain expertise to identify reagents to add or remove (Stage 1b; Supplementary Fig. 2; and Supplementary Data File 1). Cycling through 26 rounds, we tested 176 formulations identified by Design of Experiments, 116 formulations

**Table 1 | A comparison of cell-free reagent formulations**

| | Previously Published Systems | | | | | | | | | This Work | |
|---|---|---|---|---|---|---|---|---|---|---|---|
| | Jewett and Swartz: PANOx-SP[57] | Jewett and Swartz[57] | Calhoun and Swartz[65] | Zawada et al.[64] | Cai et al.[45] | Borkowski et al.[68] | Garenne et al.[21] | Warfel et al.[41] | Zhu et al[70]. | Minimal | RF$_{opt}$ |
| CFE cost ($\$_{reagent}/g_{protein}$) | $4083 | $3181 | $2803 | $339 | $564 | $3026 | $4550 | $356 | $11,433 | $175 | $60 |
| Reagent cost ($/L$_{CFE}$) | $4535 | $1958 | $1921 | $308 | $257 | $2179 | $4577 | $257 | $6974 | $93 | $143 |
| Yield determined in this work (g/L) | 1.11 ± 0.13 | 0.62 ± 0.02 | 0.69 ± 0.03 | 0.91 ± 0.08 | 0.46 ± 0.03 | 0.72 ± 0.03 | 1.01 ± 0.10 | 0.72 ± 0.05 | 0.61 ± 0.22 | 0.53 ± 0.01 | 2.39 ± 0.28 |
| Reported yield (g/L) | 0.7 | 0.7 | 0.5 | 0.7 | 1.3 | 0.8 | 4.0 | 1.2 | 1.4 | - | - |
| Mg(Glu)$_2$ | 8 | 8 | 8 | 8 | 8 | 4 | 8 | 10 | 12 | | 8 |
| NH$_4$(Glu) | 10 | 10 | 10 | 10 | | | | 10 | | | |
| K(Glu) | 130 | 130 | 130 | 130 | 260 | 80 | 80 | 130 | 130 | 300 | 362 |
| Ammonium acetate | | | | | | | | | 10 | | |
| Glucose | | | 30 | | | | | | | | 10 |
| Amino acids | 2 | 2 | 2 | 2* | 2* | 1.5 | 1.5** | 2 | 2 | 3.25 | 5 |
| Phosphate | | | 10 | 15 | 15 | | | 75 | | | 15 |
| Nicotinamide | | | | | | | | | | | 4 |
| Ribose | | | | | | | 30 | | | | 50 |
| HEPES | 57 | | | | | 50 | 50 | | 57 | | 75 |
| Bis-Tris | | | 57 | | | | | 57 | | | |
| Pyruvate | | 33 | | 35 | | | | | | | |
| Putrescine | 1 | 1 | 1 | 1 | | | | 1 | | | |
| Spermidine | 1.5 | 1.5 | 1.5 | 1.5 | 1.5 | 0.1 | 1 | 1.5 | 0.13 | | |
| Dithiothreitol | | | | | | 2.5 | 1 | | | | |
| Folinic acid | 0.03 | 0.034 | 0.034 | | | 0.035 | 0.032 | 0.034 | 2.05 | | |
| tRNA | 0.17 | 0.17 | 0.17 | | | 0.06 | 0.2 | | | | |
| CoA | 0.27 | 0.26 | 0.26 | | | 0.026 | 0.26 | | | | |
| NAD | 0.4 | 0.33 | 0.33 | | | 0.165 | 0.33 | 0.4 | | | |
| cAMP | | | | | | | 0.75 | | | | |
| PEP | 30 | | | | | | | | 40 | | |
| 3-PGA | | | | | | 9 | 30 | | | | |
| Oxalic acid | 4 | 4 | | 4 | 4 | | | 4 | | | |
| GSSG | | | | 4 | 2 | | | | | | |
| GSH | | | | 1 | | | | | | | |
| Maltodextrin | | | | | | | 21.6 | 60 | | | |
| PEG-8000 | | | | | | 2 | | | | | |
| ATP | 1.2 | 1.2 | 1.2 | | | 1.5 | 1.5 | | 2.6 | | |
| CTP | 0.85 | 0.85 | 0.85 | | | 0.9 | 0.9 | | 1.9 | | |
| GTP | 0.85 | 0.85 | 0.85 | | | 1.5 | 1.5 | | 1.9 | | |
| UTP | 0.85 | 0.85 | 0.85 | | | 0.9 | 0.9 | | 1.9 | | |
| AMP | | | | 1.2 | 1.2 | | | 1.2 | | 1.2 | 3 |
| CMP | | | | 0.86 | 0.86 | | | 0.86 | | 0.86 | 2.15 |
| GMP | | | | 0.86 | 0.86 | | | 0.86 | | 0.86 | 2.15 |
| UMP | | | | 0.86 | 0.86 | | | 0.86 | | 0.86 | 2.15 |

*1 mM tyrosine.

**1.25 mM leucine.

All numbers indicate concentration in mM, except for folinic acid, tRNA, and maltodextrin, which are given in mg/mL, PEG-8000, which is given in % w/v, and costs and yields, which have units listed.

predicted by JMP models, 131 formulations predicted by METIS, and 338 formulations designed using our domain expertise.

We found that individual additions of buffer (e.g., HEPES, Bis-Tris), magnesium glutamate, alternative energy substrates (e.g., glucose, pyruvate, maltodextrin, ribose), oxalate, spermidine, phosphate, and TCA cycle intermediates (e.g., fumarate, malate) did not improve protein yield (Supplementary Fig. 4). Moreover, individual replacements of nucleoside monophosphates—the most expensive reagent group ($62/g$_{protein}$) in the minimal formulation—with their corresponding nucleosides or nucleobases generally decreased protein yields (Supplementary Fig. 5). Insoluble precipitates were also observed when using many of these alternatives, suggesting further

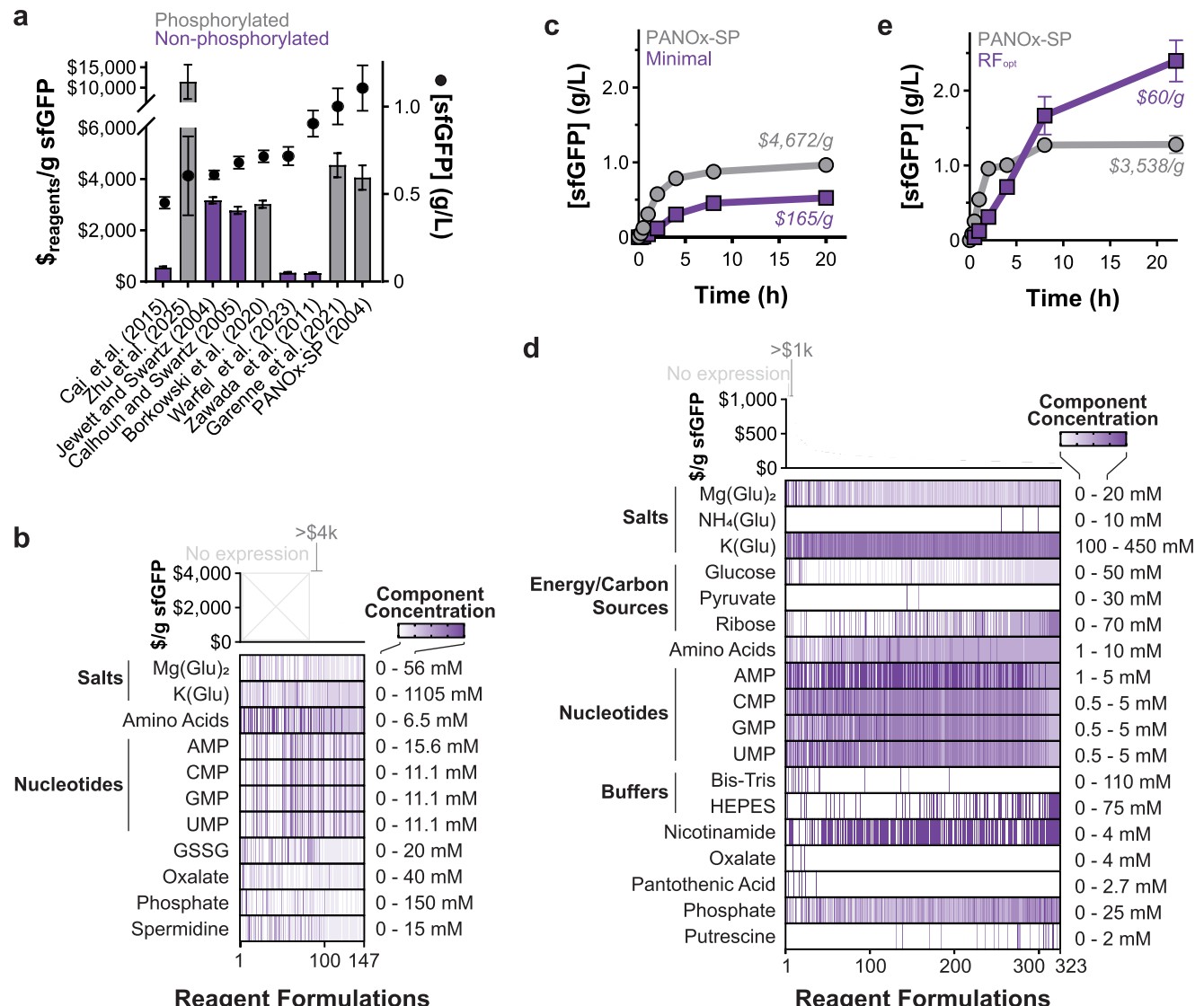

**Fig. 1 | Development of a low-cost and high-yield reagent formulation. a** sfGFP yields (dots) and cost of reagents (bars) per gram sfGFP produced were determined for 8 cell-free reagent formulations. Costs are based on only raw materials purchased at the laboratory scale, as calculated in Table 1 and Supplementary Table 2. Data are presented as mean values ± standard deviation (g/L) and cost (calculated by dividing reagent formulation cost by average sfGFP yield) ± propagated error ($/g) for n = 3 sfGFP expression replicates. Gray bars indicate formulations leveraging phosphorylated energy substrates and purple bars indicate non-phosphorylated energy substrates. **b** Reagent concentrations sorted by their corresponding $/g_protein for the initial screen of low-cost cell-free reagent components. Color gradients are used to represent component concentrations for each tested reagent formulation. **c** Cell-free expression levels over 22 h using either the PANOx-

SP or minimal reagent formulations. Reactions were performed in a 384-well plate. **d** Reagent concentrations sorted by their corresponding $/g_protein for the multi-component optimization campaign, building off the reagent formulation in (**b**). NMP concentrations were capped at 5 mM to prevent formation of insoluble precipitates. **e** Cell-free expression levels over 22 h using either the PANOx-SP or optimized reagent formulation (RF_opt). Reactions were performed in 2-mL flat-bottomed Axygen tubes. Data in panels (**c**) and (**e**) are presented as the mean values ± standard deviation of n = 3 replicates. Gray data (squares) indicates the PANOx-SP formulation and purple data (circles) indicates the RF_opt formulation. All reactions were run using BL21 Star (DE3) lysate at 30 °C for 20–22 h. Source data are provided as a Source Data file.

investigation into precipitate formation and prevention might be necessary to evaluate these modifications. Surprisingly, we found that GMP could be replaced with a combination of guanine and ribose without sacrificing protein yield and that UMP could be replaced with its corresponding base and ribose, albeit with reduced yields (Supplementary Fig. 5). These results and known synergistic benefits between reagents (e.g., glucose and phosphate[65]) suggested that any significant improvement to protein yield would require simultaneous supplementation of multiple reagents.

The best-performing formulations based on protein yield (i.e., titer) and cost per gram protein were then used as the baseline for further rounds of targeted optimization using learned design rules,

such as yield improvement through addition of ribose and HEPES. We tested 323 formulations across 10 rounds (Fig. 1d; and Stage 2; Supplementary Fig. 2; Supplementary Data File 1). Each round of optimization used the best-performing formulations by protein yield from the prior round as the new base formulation. Through this iterative process, we explored the design space of $2^{21}$ (i.e., 2,097,152) reagent combinations with only 1231 formulations, increased protein yield to over 2 g/L sfGFP, and drove down reagent costs 15-fold (Supplementary Fig. 1). We halted the optimization campaign at this point, having achieved our goal of producing protein for less than $100/g protein.

The best performing minimal reagent formulation produces 2.4 ± 0.3 g/L sfGFP within 20 h (Fig. 1e). This is an 87% increase in

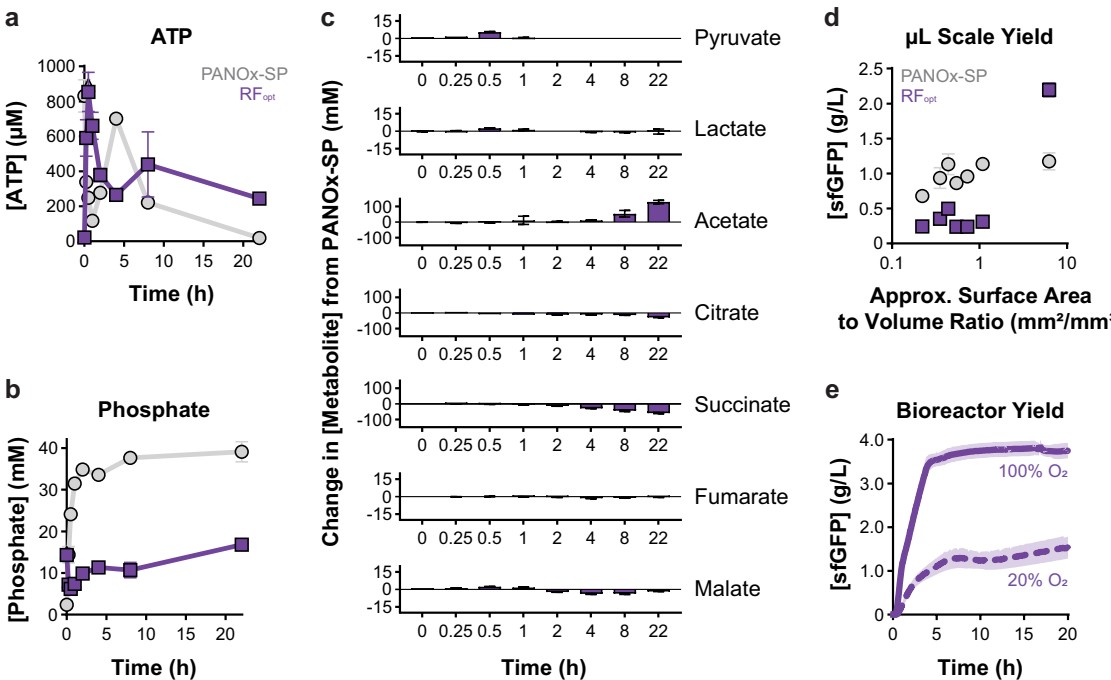

**Fig. 2 | Adjusting reagent formulations alters the cell-free metabolic environment and oxygen dependency.** **a** ATP and (**b**) phosphate concentrations in the cell-free reaction environment for the PANOx-SP and RF$_{opt}$ formulations. Data are presented as mean values ± standard deviation of $n = 3$ replicates. **c** Changes in metabolite concentration between PANOx-SP and RF$_{opt}$. Data were calculated by subtracting mean metabolite concentrations from the two formulations and are presented as mean values ± propagated error from the standard deviation of $n = 3$ replicates. **d** Comparison of sfGFP yield and approximate surface area to volume ratio (mm²/mm³), as determined by manufacturer tube/plate schematics and caliper measurements. Data are presented as mean values ± standard deviation of $n = 3$ replicates. **e** sfGFP production in a 4-mL bioreactor supplemented with either a 20% or 100% $O_2$ feed. Data are presented as mean values ± standard deviation of $n = 6$ bioreactors with a 20% $O_2$ feed and $n = 3$ bioreactors with a 100% $O_2$ feed. Additional replicates were run for the 20% $O_2$ feed due to the increased variability. All reactions were run using *E. coli* BL21 Star (DE3) lysate at 30 °C for 20-22 h. In **a**, **b** and **d**, gray data points (circles) represent the PANOx-SP formulation and purple data points (squares) represent the RF$_{opt}$ formulation. Source data are provided as a Source Data file.

protein yield over the previously best-performing PANOx-SP system. The optimized reagent formulation (RF$_{opt}$) costs $143/L$_{CFE}$ ($60/g$_{protein}$), 99% less than the PANOx-SP system and 84% less than the previous lowest cost system—a more than one order of magnitude cost reduction (Table 1). The optimization campaign required approximately 175 h of wet-lab work, 37 mL of cell-free reactions across the 1,231 formulations tested in triplicate, and only $8.35 for all small molecule reagents.

## Characterizing the optimized reagent formulation
To better understand the behavior of our optimized minimal reagent formulation, we compared cell-free metabolism over the course of a 20-h cell-free expression reaction using the PANOx-SP formulation and our optimized reagent formulation (RF$_{opt}$). We first observed ATP and phosphate profiles as these have been shown to be key measures of cell-free gene expression performance[32,71,72]. In line with previous works using the PANOx-SP formulation[71], a surge of ATP is produced within 30 min with a corresponding rapid and sustained increase in phosphate as a result of the phosphorylated energy substrate PEP being consumed (Fig. 2a, b). ATP levels then drop to nearly zero by the end of the reaction. RF$_{opt}$ generates comparable levels of ATP within 30 min, despite starting with no exogenous ATP, and sustains ATP levels at ~250 μM (well above the ATP affinity threshold for cell-free gene expression[72]) for the duration of the reaction (Fig. 2a). Phosphate levels hover around 10-15 mM over the reaction lifetime, which is three times lower than the PANOx-SP formulation because free phosphate is not liberated from PEP (Fig. 2b). Furthermore, we measured key central carbon metabolites (pyruvate, lactate, acetate, citrate, succinate, fumarate, and malate) that often serve as carbon and energy intermediates and sinks in metabolism for both formulations[65,73,74] (Fig. 3c;

and Supplementary Fig. 6). Time-course measurements show that RF$_{opt}$ sustains the tricarboxylic acid cycle metabolites at much lower concentrations during the reaction compared to the PANOx-SP formulation. These results suggest that RF$_{opt}$ balances cell-free metabolism for a longer period, allowing for more sustained rates of protein synthesis.

Next, we investigated the performance of RF$_{opt}$ across different reaction geometries, known to be important for oxygen transfer[75,76]. To do this we tested several reaction vessels (i.e., 2-mL tubes, 0.2-mL tubes, 384-well plates) and volumes (i.e., 15-100 μL) to vary the surface area-to-volume ratio between 0.1 and 10 (Fig. 2d; and Supplementary Fig. 7). We found that cell-free expression using the optimized formulation had greater sensitivity to this ratio, suggesting oxygen levels may limit reaction yields. This finding supports previous demonstrations that oxidative phosphorylation is active in crude cell-free lysates and required for sufficient energy regeneration[75]. We leveraged a 4-mL membrane-based bioreactor with a tunable oxygen feed[77] to further examine this behavior (Supplementary Fig. 8). When provided with a 20% $O_2$ feed to mimic ambient air, the bioreactor-housed cell-free gene expression reaction produced less than 2 g/L sfGFP (Fig. 2e). Shifting to a 100% $O_2$ feed increased the dissolved oxygen (dO$_2$) content in the reaction (Supplementary Fig. 8) and improved sfGFP production by 45%, achieving $3.7 ± 0.2$ g/L sfGFP ($39/g$_{protein}$) at the 4-mL scale (Fig. 2e). Protein synthesis is also more rapid, producing $0.9 ± 0.1$ g/L/h for the first four h in the bioreactor. In contrast, the 20% $O_2$ feed produced roughly $0.3 ± 0.04$ g/L/h and the 15-μL tube-based reaction produced $0.2 ± 0.01$ g/L/h. These results suggest that control of metabolism and oxygen transfer will be key to extending reaction rate, longevity, and yield.

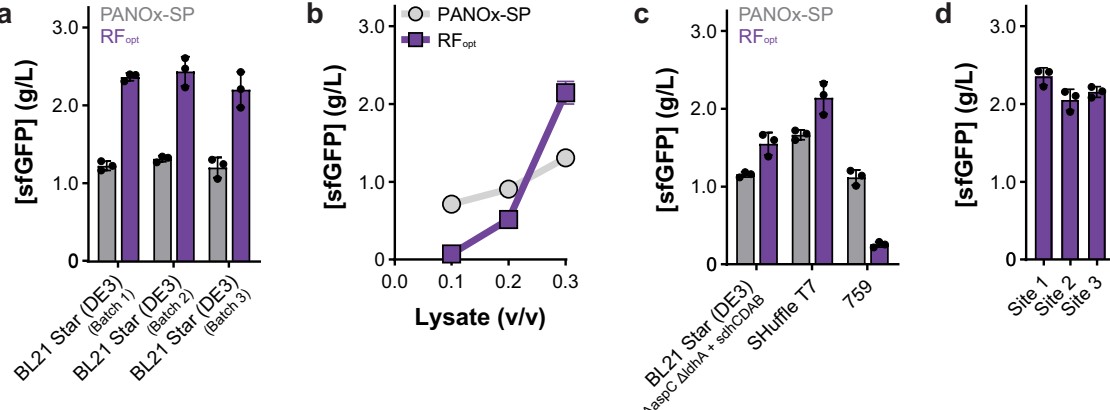

Fig. 3 | The optimized reagent formulation (RF_opt) remains robust across batches of cell lysates and users/locations. a Productivity of the PANOx-SP and RF_opt formulations across three independent *E. coli* BL21 Star (DE3) lysate batches. b Impact of lysate concentration on sfGFP expression level. New optimum magnesium and phosphate concentrations were determined for each lysate concentration. c sfGFP expression in three additional *E. coli* lysates with optimized magnesium and phosphate concentrations. d Protein production using RF_opt at three different laboratory sites, each using reagents and lysate prepared independently. In all panels, data are presented as mean values ± standard deviation of $n = 3$ replicates. Gray data bars or points represent the PANOx-SP formulation and purple data bars or points represent the RF_opt formulation. All reactions were run at 30 °C for 20–22 h. Source data are provided as a Source Data file.

## Evaluating robustness of an optimized reagent formulation

Cell-free reagent formulations must be robust to failure across batches of cell lysates, users/locations, and in the synthesis of different proteins for widespread use. To test robustness, we performed a series of experiments measuring sensitivity to these important parameters. We evaluated our optimized reagent formulation (RF_opt) using various cell lysate batches, volumes, and source strains. First, we grew three independent cultures of *E. coli* BL21 Star (DE3) and made subsequent lysates as described in Methods. Magnesium and phosphate concentrations were optimized for each new lysate (Supplementary Fig. 9). Running cell-free expression reactions using these lysates and RF_opt produces consistent protein yields across these three lysate batches (Fig. 3a). Next, we found that protein yields are more sensitive to the amount of lysate added than using the PANOx-SP formulation (Fig. 3b), reiterating the fine metabolic balance we observed (Fig. 2). We then assessed RF_opt for cell-free expression with lysates made from other strains of *E. coli*. We specifically evaluated three other strains: (i) a genomically modified variant of BL21 Star (DE3) to test sensitivity to typical metabolic modifications; (ii) SHuffle T7, another B-strain commonly used with enhanced capacity to correctly fold proteins with disulfide bonds; and (iii) a K-strain *E. coli*, 759, engineered for expanded genetic code applications[24,78]. Following lysate preparation, we ran cell-free expression reactions with RF_opt (Fig. 3c; and Supplementary Fig. 9). We found that RF_opt significantly improves protein yields in B-strain-derived lysates compared to the PANOx-SP formulation, whereas RF_opt decreases protein yields in a K-strain derived lysate, requiring additional optimization as seen before[24,78]. Although it is unclear whether RF_opt can be applied to other bacterial or eukaryotic lysates directly, recent work suggests that some level of transfer learning may be achievable[70,79]. We also found that RF_opt is robust to the addition of common salts, osmolytes, and protein buffers (Supplementary Fig. 10), enabling use of the formulation across a variety of workflows.

In addition to a reagent formulation needing to be robust across batches of cell lysates, cross-laboratory reproducibility is essential for wider adoption of cell-free gene expression. While multi-site implementation has proven challenging[80,81], we wanted to test RF_opt across different locations. To do this, we provided the RF_opt formulation to two other academic institutions, with a different researcher preparing reagents and setting up reactions at each site. We found that RF_opt produced protein yields within ~9% between sites (Fig. 3d), demonstrating that this formulation is robust across users/locations.

Lastly, we sought to address potential reporter bias that stems from using sfGFP as the reporter protein during the optimization campaign. sfGFP is highly stable and simple to express and fold[82]. We therefore wanted to ensure that RF_opt could be used effectively to synthesize a variety of more complex, commercially relevant protein products. We first chose to produce FDA-approved carrier proteins (protein D and CRM197) used in conjugate vaccines as well as the trastuzumab monoclonal antibody (both Fc domain and full-length antibody) used to treat HER2-positive breast and stomach cancers as model proteins of interest. While RF_opt produced some soluble product, it was less productive than the PANOx-SP formulation. However, neither condition produced the trastuzumab Fc dimer because oxidizing conditions are required to form disulfide bonds (Fig. 4a; and Supplementary Fig. 11).

To form disulfide bonds, cell-free reactions use pretreatment with iodoacetamide (IAM) to deactivate reductases, addition of glutathione (GSSG/GSH) to create an oxidizing reaction environment, and supplementation of disulfide bond isomerase (e.g., DsbC) to aid in disulfide bond shuffling[47,83,84]. Surprisingly, the addition of these components to RF_opt significantly reduced sfGFP production by 75-97% (Fig. 4b). To mitigate indiscriminate IAM activity[85], we knocked out a glutathione reductase gene (Δ*gor*) in the lysate source strain to help stabilize the oxidizing lysate environment, which recovered protein yields with 20-fold less IAM (Fig. 4b; and Supplementary Fig. 12). However, adding glutathione at several oxidized:reduced ratios still decreased protein yields by up to 62% (Fig. 4b; and Supplementary Fig. 13), without negatively impacting cell-free energy generation (Supplementary Fig. 14). Diving deeper, we found that glutathione addition correlated with a decrease in reaction pH for both the RF_opt and the PANOx-SP formulations (Fig. 4c). While this was not problematic for the PANOx-SP formulation, which remained within the cytoplasmic pH of 7.4-7.8[86], RF_opt was more acidic and the addition of glutathione rapidly dropped the reaction pH below pH 7.0 (Fig. 4c). By increasing the HEPES buffer pH to 7.5, we avoided the pH drop and recovered protein yield (Fig. 4c; and Supplementary Fig. 15).

The oxidized version of RF_opt improved production of protein D, CRM197, and the trastuzumab Fc and full-length antibody constructs (Fig. 4d), increasing the fraction of assembled Fc domains and full-length antibodies by 44% and 90%, respectively, compared to an oxidized PANOx-SP reaction (Supplementary Fig. 16). We then examined the preliminary IgG1 light chain expression to further improve full-length trastuzumab folding. By lowering the reaction temperature[76,87]

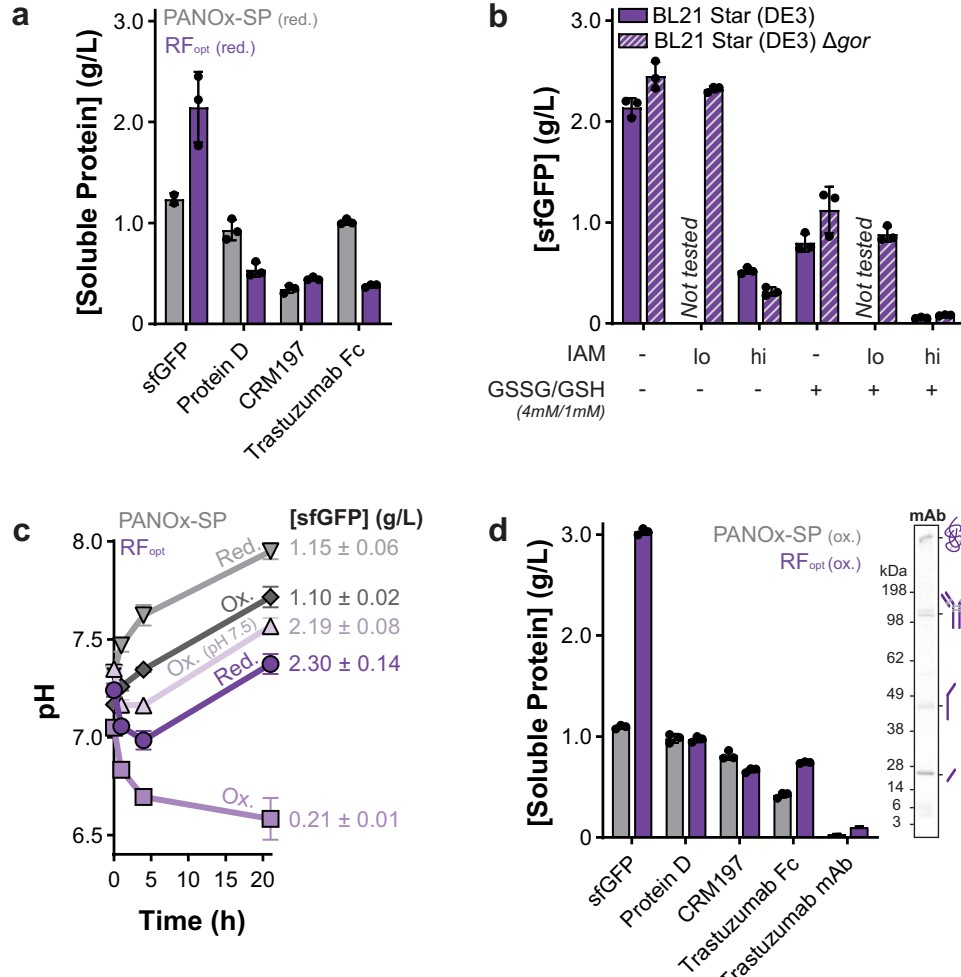

**Fig. 4 | An oxidized variation of the optimized reagent formulation (RF$_{opt}$) produces full-length aglycosylated antibody. a** Soluble protein production in the unmodified PANOx-SP and RF$_{opt}$ formulations using a BL21 Star (DE3) lysate. **b** Impact of oxidized/reduced glutathione (GSSG/GSH), iodoacetamide (IAM; lo = 25 μM, hi = 500 μM), and Δ*gor* genome modification (striped bars) on sfGFP yields when using RF$_{opt}$. **c** pH traces and corresponding yields for a 21-h reaction when using the PANOx-SP or RF$_{opt}$ formulations either in their unmodified reducing condition (red.) or with the addition of IAM, GSSG, and GSH (ox.). Unless otherwise noted, the HEPES buffer included in each reaction was initially at pH 7.2. **d** Soluble protein production in the oxidized versions of the PANOx-SP and RF$_{opt}$ formulations. The insert shows the autoradiogram for trastuzumab full-length antibody (mAb) formation in the RF$_{opt}$ system. A BL21 Star (DE3) Δ*gor* lysate was used in panels (**c**, **d**). All reactions were run at 30 °C for 20–22 h. Data are presented as mean values ± standard deviation of *n* = 3 replicates. Gray bars or points represent the PANOx-SP formulation and purple bars or points represent the RF$_{opt}$ formulation. Source data are provided as a Source Data file.

from 30 °C to 22 °C and delaying the addition of DNA encoding the heavy chain[46] for 3-h, we produced ~150 μg/mL full-length trastuzumab using RF$_{opt}$ (Supplementary Fig. 17).

**Producing diverse proteins in an optimized formulation**

We next expressed fifteen medically relevant recombinant proteins and compared product yields to the most productive (PANOx-SP) and lowest $/g sfGFP cost reagent formulations[64] (Fig. 5a). The selected proteins span 16-147 kDa, 0-16 disulfide bonds, and a variety of clinical applications, with six previously expressed in *E. coli*-based cell-free expression systems[25,40,42,52,84]. All protein yields were determined via [14]C-leucine incorporation in an oxidized cell-free reaction environment with added bacterial DsbC and verified with autoradiography (Supplementary Fig. 18–21). We observed expression of all 15 therapeutic proteins with our optimized reagent formulation (RF$_{opt}$) and obtained >100 μg/mL soluble full-length protein for 12 proteins with at least one formulation. RF$_{opt}$ yielded the same or more soluble protein than the previous most productive (PANOx-SP) formulation for 13 of 15 proteins and had lower $/g$_{protein}$ costs for 11 of 15 proteins than the previous lowest $/g$_{protein}$ cost formulation[64] (Supplementary Fig. 22). However,

the level of improvement varied across proteins. For instance, caplacizumab was expressed with significantly higher yields using RF$_{opt}$, while streptokinase and myoglobin showed reduced yields compared to the PANOx-SP formulation (Fig. 5a). In addition, while the ratio of antibody heavy chain and light chain varied across reagent formulation, the non-phosphorylated energy systems reduced insoluble aggregation of antibody constructs (Supplementary Fig. 23). We further demonstrated expression for seven diverse, non-therapeutic proteins in RF$_{opt}$, with 4 of 7 proteins achieving statistically higher soluble expression levels in RF$_{opt}$ (Supplementary Fig. 24). Looking holistically, all proteins had lower $/g$_{protein}$ costs in the RF$_{opt}$ formulation than the PANOx-SP formulation.

Finally, we wanted to show that proteins expressed with RF$_{opt}$ were functional. We selected a subset of our expressed proteins with a variety of functions—vtPA, colicin M, TRI2-2, and trastuzumab—to test for protein activity. vtPA is an enzymatically active truncated form of tissue plasminogen activator, which cleaves an arginine-valine peptide bond in plasminogen to activate the protease and dissolve blood clots[84,88]. We found that cell-free-expressed vtPA is functional and able to rapidly cleave the fluorescent 7-amino-4-methylcoumarin (AMC)

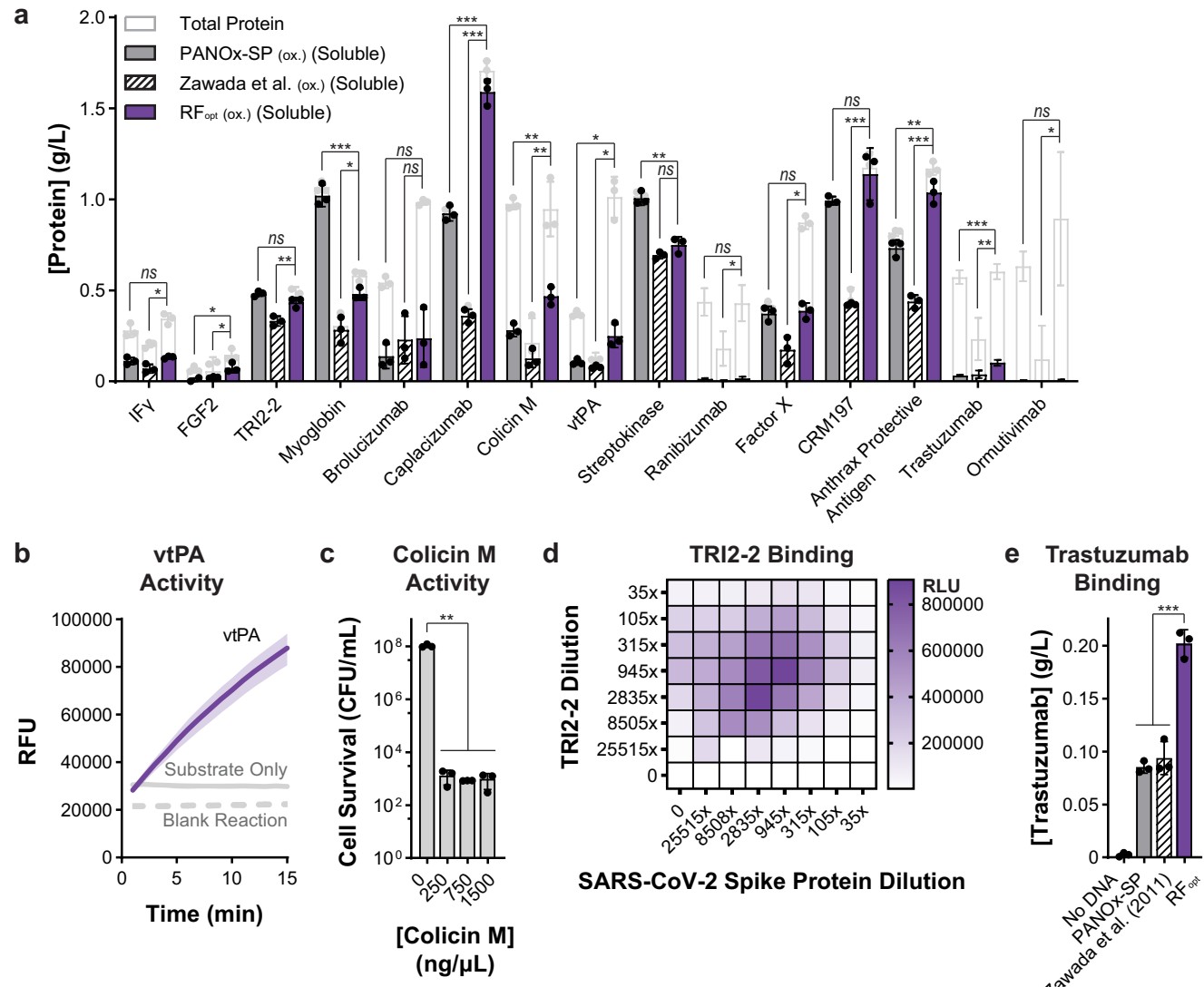

**Fig. 5 | The optimized reagent system (RF_opt) expresses 15 diverse recombinant protein therapeutics. a** Expression of 15 recombinant protein therapeutics using three different reagent formulations, as quantified by $^{14}$C-leucine incorporation. Both total and soluble protein yields are shown for n = 3 replicates and data are presented as mean values ± standard deviation. Statistical significance for soluble yields was calculated by unpaired two-tailed t-tests (adjusted $p$ value < 0.001 denoted by ***, 0.001 to 0.01 by **, 0.01 to 0.05 by *, and > 0.05 by ns). White bars represent total protein and colored bars represent soluble protein from PANOx-SP (gray), Zawada et al. (2011) (striped), and RF_opt (purple) formulations. **b** Enzymatic activity of vtPA produced using RF_opt. Data are presented as mean values ± standard deviation of $n = 3$ replicates. The blank reaction contained all cell-free expression components except a DNA template. **c** Viability of CLM24 indicator cells treated with colicin M produced using RF_opt. Data are presented as mean values ± standard deviation of $n = 3$ independent cell cultures. Statistical significance was calculated by an unpaired two-tailed t-test ($p = 0.007$). **d** AlphaLISA binding pattern generated by interaction of cell-free produced TRI2-2 minibinder and SARS-CoV-2 spike protein. This experiment was run in duplicate; all results are in Supplementary Fig. 23. "0" indicates AlphaLISA reactions that do not contain any of the respective reaction component. (e) Full-length trastuzumab detected using an anti-idiotypic sandwich ELISA. Data are presented as mean values ± standard deviation of $n = 3$ replicates. Statistical significance was calculated by an unpaired two-tailed t-test ($p = 0.0001$ for PANOx-SP and $p = 0.0008$ for Zawada, et al. (2011)). Cell-free reactions used to generate protein for panels (**a**–**e**) were run in an oxidizing cell-free environment at 30 °C for 20 h using BL21 Star (DE3) Δ*gor* lysate supplemented with 5 µM purified bacterial DsbC. For antibody production, heavy chain pDNA was added after 1.5 h of incubation. Source data for all figures and corresponding $p$ values for Fig. 5a, c, and e are provided as a Source Data file.

from a short Ile-Pro-Arg peptide sequence[89] (Fig. 5b). Next, we tested antimicrobial activity of colicin M[90] by incubating cell-free-expressed colicin M with *E. coli* CLM24 indicator cells. The presence of colicin M decreased *E. coli* colony formation by a factor of 10$^5$ compared to a no-DNA cell-free reaction matrix (Fig. 5c). Lastly, we tested TRI2-2 and trastuzumab function through binding assays. Using AlphaLISA—a protein-protein interaction assay that produces luminescence based on the proximity of donor and acceptor beads decorated with the proteins of interest[91]—we found cell-free-expressed TRI2-2, a computationally designed miniprotein inhibitor of SARS-CoV-2[25], indeed binds to the SARS-CoV-2 spike protein to produce AlphaLISA signal

2.4-fold over background (Fig. 5d, and Supplementary Fig. 25). With an anti-idiotypic sandwich ELISA assay—trastuzumab antibody bound to a capture antibody designed to interact with the trastuzumab antigen binding site[92]—cell-free-expressed trastuzumab effectively bound to the capture antibody (Fig. 5e). These results demonstrate that RF_opt can robustly express active proteins with a variety of functions.

## Discussion

Cost and productivity are pivotal determinants for widespread use of cell-free systems for protein production. Here, we explored 1,231 reaction mixtures to establish a low-cost, high-yielding cell-free

reagent formulation capable of producing > 2 g/L protein product for $143/$L_{CFE}$ in reagents. We identified a minimal reagent formulation containing only six components (with all 20 amino acids counted as one component), which comprises the fewest components of any reported formulation, to our knowledge. We also developed an optimized minimal reagent formulation with 12 components. This optimized formulation ($RF_{opt}$) reduces reagent cost from 75% of total material costs to less than 10% (Supplementary Table 1).

The optimized reagent formulation ($RF_{opt}$) has several key features. First, it has the highest reported protein yield for a reagent mixture leveraging non-phosphorylated energy substrates and the lowest reported cost to date (~$60/$g_{protein}$). We anticipate that economies of scale will further decrease cost for industrial applications, with techno-economic analyses of cell-free platforms suggesting bulk reagent costs may be 10% or less of the laboratory scale cost[93,94]. A notable discovery was the importance of dissolved oxygen concentration on system performance; high $DO_2$ levels increased yields to > 3.7 g/L in a bioreactor, approaching the highest reported yields in a cell-free system (4 g/L)[60], but for a fraction of the cost (~$39/$g_{protein}$). Second, the formulation is robust to failure across batches of cell lysates, users/locations, and in the synthesis of different proteins. Third, the platform can produce proteins with disulfide bonds when pH is controlled. These features increase the economic viability of using cell-free expression for protein design[23,25,27], protein engineering[22,26,95], and therapeutic production[40,42,44,45].

$RF_{opt}$ outperforms previously optimized reagent formulations for *E. coli* cell-free expression, including formulations identified with machine learning[70] or active learning-guided[68] optimization strategies for the same or similar *E. coli* lysates. We found that adding and removing reagent components across our exploration, rather than optimizing a fixed set of components, was key to designing $RF_{opt}$. By blending human intelligence (domain knowledge) and computational reaction design, we were able to more effectively optimize reagent formulations. While we did not find a single computational design tool to be superior, relying solely on computational predictions and a single, defined reagent set is not sufficient for achieving high yields. A combination of computer-guided optimization followed by targeted additive exploration based on project goals may be more accessible to laboratories without access to domain knowledge.

This work highlights the modular nature of cell-free systems, which allows for on-demand production of a diverse product library. Importantly, by just changing the DNA input, the same system formulation can be used to make different protein products. We showcased the optimized reagent formulation by synthesizing fifteen therapeutic proteins spanning 16-147 kDa, 0-16 disulfide bonds, and a variety of clinical applications (Fig. 5). Nine of these proteins had never been expressed in cell-free systems, and 12 proteins were produced at > 100 µg/mL soluble full-length protein. Across all proteins for which soluble expression levels (g/L) were quantified, 11 of 23 had higher expression levels with the $RF_{opt}$ system, 10 of 23 were statistically the same, and 2 of 23 performed worse. However, all 23 proteins performed better with $RF_{opt}$ when compared on a reagent cost per gram protein produced basis. Variability in overall expression yield may be linked to DNA design, optimal pH values, or folding rates.

With these yields and costs, cell-free systems could feasibly be integrated into point-of-care or distributed biologics manufacturing paradigms to complement centralized facilities that rely on cellular protein expression. Additional formulation optimization, such as through the addition of chaperones[96,97], metabolic proteins[73,98], crowding agents[99,100], and vesicles or nanodiscs[101], may help to improve production of complex, large, or uncommon therapeutic proteins, or those with structural modifications. Coupled with the large body of work on freeze-dried systems[41,49,102], cell-free systems provide the necessary flexibility and reduction in transportation and storage costs for distributed, on-demand manufacturing[103–106].

While our work brings us a step closer to realizing the vast potential of cell-free systems in distributed manufacturing, continued advances in purification strategies[44,107–109] are required. Further reductions in cost, especially involving DNA template preparation[76,110], nucleoside monophosphates, and cell lysate, are needed, and increases in productivity and longevity are also necessary. Our work highlights one potential path. Specifically, we demonstrated with our minimal formulation that a combination of guanine and ribose can replace GMP without a decrease in system performance (Supplementary Fig. 5). These findings indicate that the metabolic pathways necessary to build and sustain full reagent profiles from a limited number of low-cost substrates are active and can be intentionally designed. We also demonstrate with the $RF_{opt}$ formulation that neither inhibitory phosphate concentrations nor ATP depletion are responsible for termination of cell-free gene expression in our lysates, suggesting that optimizing the system for stability of other small molecules, RNA, or cellular machinery may be necessary to extend reaction duration. Combining these results with advances in self-sustaining translation machinery[111,112], DNA templates[113], and amino acids[114] may provide a path to a lower-cost and longer-lasting cell-free platform capable of building all components it needs to function.

In sum, the low-cost, high-yielding cell-free platform developed in this work offers a robust and versatile approach to rapidly produce recombinant proteins. By doubling yields of conventional cell-free technologies, we anticipate that the optimized reagent formulation will be adopted across numerous application spaces, from supporting protein design efforts to cell-free biomanufacturing at industrial scales for biologics production.

## Methods

### Cell extract preparation

Crude extracts were made from multiple *Escherichia coli* strains, listed in Supplementary Table 3, based on past protocols[57,115,116]. In brief, cells were grown overnight at 37 °C in LB media (10 g/L tryptone, Sigma-Aldrich T7293; 5 g/L yeast extract, Thermo Scientific 212720; 5 g/L NaCl, Sigma-Aldrich S3014) to prepare a starter culture. The following day, 1 L cultures of sterilized 2xYTPG media (16 g/L tryptone; 10 g/L yeast extract; 5 g/L NaCl; 7 g/L $K_2HPO_4$, Sigma-Aldrich 60353; 3 g/L $KH_2PO_4$, Sigma-Aldrich P0662; and 18 g/L glucose, Millipore Sigma 70990, adjusted to a pH of 7.2 with KOH, Sigma-Aldrich P1767) were inoculated with the overnight culture to an initial $OD_{600}$ of 0.06-0.08 in Tunair shake flasks. Cells were grown at 37 °C and 250 rpm to an $OD_{600}$ of 0.6 and inoculated with 0.5 mM isopropyl-β-D-thiogalacto-pyranoside (IPTG, Sigma-Aldrich I6758) to induce T7 RNA polymerase expression. At an $OD_{600}$ of 3.0, cells were harvested by centrifugation at 5,000 g for 10 min at 4 °C. The resulting cell pellets were then resuspended with S30 buffer (10 mM Tris acetate pH 8.2, Sigma-Aldrich 93337; 14 mM magnesium acetate, Sigma-Aldrich M5661; and 60 mM potassium acetate, Sigma-Aldrich P1190) and pelleted by centrifugation at 10,000 g for 2 min. Cells were washed a total of three times. After the final centrifugation step, the cell pellet mass was recorded, and the cells were flash frozen in liquid nitrogen and stored at -80 °C.

Frozen cells were thawed on ice for 60 min and resuspended in 1 mL/g S30 buffer. Cells were then lysed with a single pass at 20,000-25,000 psi through either an Avestin EmusliFlex B15 or C3 homogenizer. The resulting lysate was centrifuged at 12,000 g for 10 min and the supernatant collected. This step was performed twice. The final clarified lysate was aliquoted, flash frozen in liquid nitrogen, and stored at -80 °C.

### Lysate source strain engineering

The BL21 Star (DE3) Δgor strain was constructed using the pcrEG and pEcCpfIH plasmids as previously described[117]. Initially, pEcCpfIH was introduced into BL21 Star (DE3) by chemical transformation. The strain containing the editing plasmid was utilized to prepare electro-

competent cells and was induced with 50 mM arabinose (GoldBio A-300-500) at $OD_{600}$ 0.1 and harvested at $OD_{600}$ 0.6. Cells were washed with two washes of water and two washes of 10% w/v glycerol (Sigma-Aldrich G5516).

Concurrently, golden gate assembly was used to construct the cRNA expression plasmid, using the CRISPOR[118] designed guide ATAGGAAGTATGAATACGGTCGA, targeting the *gor* gene. Homology directed repair (HDR) templates were designed containing 45 bp up- and downstream of the *gor* gene and ordered with phosphorothioate modified ends. Both the pcrEG-cRNA plasmid and the HDR templates were transformed into the prepared BL21 Star (DE3) strain containing the editing plasmid and recovered for 2-3 h at 37 °C before selection on LB agar containing 50 μg mL$^{-1}$ kanamycin (Sigma-Aldrich K4000) and 100 μg mL$^{-1}$ spectinomycin (Sigma-Aldrich S4014).

Once colonies were confirmed for whole gene removal by colony PCR using Q5 hot start polymerase (NEB) and the primers gor-up-F 5′-ATT GAACTGGCGGTACTGCC-3′ and gor-down-R 5′-GTCA-GAAGTACGGGTGGTGC-3′, the pcrEG-cRNA plasmid was removed by growing in 10 mM rhamnose overnight and streaked onto LB (no antibiotic) plates. Subsequently, the pEcCpfIH plasmid was removed by growing the strains overnight in LB containing 5 g L$^{-1}$ glucose and streaked onto LB agar containing 5 g L$^{-1}$ glucose and 10 g L$^{-1}$ sucrose (Sigma-Aldrich S0389).

Bacterial genome sequencing was performed by Plasmidsaurus using Oxford Nanopore Technology with custom analysis and annotation.

### DNA template preparation

All protein sequences used in this paper are listed in Supplementary Table 4. Signal sequences and pro-peptides were removed when present to leave only the final, activated sequence. Unless otherwise noted, gene sequences were codon optimized for expression in *E. coli* and synthesized into a pJL1 backbone at the NdeI/SalI restriction sites by Twist Biosciences. SnapGene 5.0.8 was used to confirm plasmid designs. Plasmids were purified for use in cell-free expression reactions with the Qiagen HiSpeed Plasmid Midi Kit (Qiagen 12643) and further cleaned with an ethanol precipitation.

### Protein Purification

A plasmid containing DsbC with a 5′ CAT-Strep-linker (CSL) tag was transformed into NEB BL21(DE3) Competent *E. coli* cells (NEB C2527H) and plated on LB agar containing 50 μg/mL kanamycin. The following day, a 50 mL culture of Overnight Express™ Instant TB Media (EMD Millipore 71491-4) was inoculated with a single colony of transformed BL21(DE3) and grown overnight at 37 °C and 250 rpm. Cells were pelleted at 4,000 g and 4 °C for 20 min. and resuspended in 2 mL/g of BugBuster® Master Mix (EMD Millipore 71456-3). After incubation at room temperature for 15 min., the lysed cells were centrifuged at 10,000 g for 10 min. to remove insoluble components. Meanwhile, 1 mL of Strep-Tactin®XT 4Flow® high-capacity resin (IBA 2-5010-002) was loaded onto a polypropylene column (Bio-Rad) and equilibrated with 2 column volumes of Buffer W (IBA 2-1003-100). The lysed cell supernatant was added to the column and washed five times with 1 column volume of Buffer W. Protein was eluted with Buffer BXT (IBA 2-1042-025). The most concentrated elution fractions were pooled and dialyzed into a buffer containing 100 mM Tris-Cl (Sigma-Aldrich T5941) and 150 mM NaCl at pH 8. Protein yield and purity were assessed with a Bradford assay (Bio-Rad 5000205 using an Agilent BioTek Synergy H1 plate reader and Gen5 version 3.14) and SDS-PAGE gel (Fisher scientific NP0322BOX). The purified protein was flash frozen in liquid nitrogen and stored at -80 °C.

### Cell-free gene expression reactions

10-15-μL cell-free reactions were performed at 30 °C in 2-mL microcentrifuge tubes (Axygen MCT-200-A) for 20 h. All reactions contained 13.3 ng/μL plasmid and 30% v/v crude cell extract, unless otherwise noted. Reagent mixture compositions are detailed in Table 1; catalog information and preparation notes are provided in Supplementary Table 2. Reagents were thawed on ice and combined at room temperature to prevent precipitation of NMPs, which were always added to the reagent mixture last. To create an oxidizing reaction environment, cell extracts were treated with 500 μM or 25 μM iodoacetamide (IAM) at room temperature for 30 min. before use. An additional 4 mM oxidized glutathione (GSSG) and 1 mM reduced glutathione (GSH) were added to the reaction mixture. In some noted cases, 5 μM purified DsbC was also added to the reaction mixture[83,119].

### Bioreactor tests

Custom, lab-made bioreactors[77] were rinsed with 70% ethanol, followed by deionized water, air-dried, assembled, and autoclaved with a 45-min hold at 121 °C. After autoclaving, the reactors were cooled to 4 °C for approximately 20 min. and then warmed to 30 °C in the reactor incubator for approximately 30 min. The components of the cell-free reagent mixture were thawed on ice, combined within a biosafety cabinet, and thoroughly vortexed between the addition of individual reagents. The reagent mixture and crude cell lysate were measured and aliquoted from a master mix container into 1.5-mL tubes for each of the three reactors and kept on ice until use. Dissolved oxygen and sfGFP fluorescence sensors were calibrated, and the reactors were flushed with the specified gas for approximately 10 min. prior to initiating the reactions. Cell-free reactions were initiated by drawing 2800 μL of the reagent mix into a 5-mL syringe equipped with a blunt-tip 18-gauge needle and subsequently introduced into the reactor via a feed line. The impeller and sensors were activated as 1200 μL of crude cell lysate was introduced into the reactor using a 3 mL syringe. The reactions were conducted for 20 h with a humidified gas flow of 400 cubic centimeters per minute into the reactor jacket, maintaining a constant impeller speed of 500 rpm and a temperature of 30 °C. After 20 h, the final cell-free reaction was harvested through the feed line, and the final reaction volume and sfGFP concentrations were assessed using a fluorescein standard.

### Protein expression quantification

To assess the amount of sfGFP production, 2 μL of each cell-free expression (CFE) reaction were diluted with 48 μL of nanopure water in a black Corning Costar 96-well flat-bottom plate (Costar 3694). Fluorescence was read with 485 nm excitation and 528 nm emission using an Agilent BioTek Synergy H1 plate reader and Gen5 version 3.14 software, with values converted to sfGFP concentration via a standard curve derived from sfGFP measured using $^{14}C$-leucine incorporation.

All other proteins were quantified using radioactivity, based on previously developed methods[120]. Briefly, 10 μM $^{14}C$-leucine (PerkinElmer NEC279E001MC) was included in standard CFE reactions. After incubation, 5 μL of the total CFE reaction was treated with 100 μL 0.1 N KOH and incubated at 37 °C for 20 min. The remaining CFE reaction was centrifuged at 16,100 g for 10 min, and 5 μL of the soluble supernatant was treated with KOH as well. 50 μL aliquots of the treated reactions were spotted onto two strips of Whatman 3MM CHR cellulose chromatography paper (WHA3001614) and dried under a heat lamp. One of the two chromatography paper strips was then placed in a beaker and washed three times with 5% w/v trichloroacetic acid (Sigma-Aldrich T6399) for 15 min at 4 °C, followed by a wash with 200 proof ethanol (Sigma-Aldrich E7023) at room temperature. The washed paper strips were dried under a heat lamp. Radioactivity was then measured with a PerkinElmer MicroBeta2 (MicroBeta2 Windows Workstation version 6.0.0.0) with CytoScint liquid scintillation cocktail.

Autoradiograms were developed by separating total fractions of CFE reactions containing 10 μM $^{14}C$-leucine via SDS-PAGE. To visualize

disulfide bond formation, samples were not denatured before analysis. Otherwise, samples were treated with dithiothreitol and denatured at 70 °C for 3 min. SDS-PAGE gels were vacuum-dried between two cellophane sheets with a Hoefer slab gel dryer and exposed to a phosphor screen for at least three days to generate autoradiographs. Autoradiogram gels were imaged with a GE Typhoon 7000 (Typhoon FLA 7000 control software version 1.2.1.93). Protein bands were analyzed with densitometry (ImageJ 0.5.8).

Total yield of monoclonal antibodies was calculated by first calculating total yield from the [14]C-leucine incorporation data assuming 100% heavy chain and 100% light chain production. These values were then averaged, due to the similar ratio of molecular weight to number of leucines. Full-length antibody yields were then calculated by multiplying this total yield by the percent full-length antibody calculated via densitometry from the oxidizing SDS-PAGE gels. Error was propagated through each step and visualized in plots with the error bars.

## Protein activity assays

vtPA activity was determined by diluting cell-free reactions containing expressed vtPA 1:10 in PBS and incubating with 25.5 μM D-Ile-Pro-Arg-AMC peptide (iPR-AMC, Echelon Biosciences 855-18). Peptide cleavage by vtPA was determined by release of the free fluorescent 7-amino-4-methylcoumarin (AMC) over the course of an h at 26 °C and measured via a BioTek Neo2 plate reader with 354 nm excitation and 442 nm emission.

Colicin M antimicrobial activity was determined through incubation with CLM24 indicator cells, as previously described[52]. An overnight culture of LB media inoculated with CLM24 cells was diluted 1:100 in fresh LB media. The CLM24 culture was incubated at 37 °C and 220 rpm to an $OD_{600}$ of 0.7–0.9. The cells were then pelleted at 3000 $g$ for 5 min., washed twice with 0.85% w/v sodium chloride, and resuspended in LB media to an $OD_{600}$ of 0.1. Cell-free reactions containing diluted colicin M were added to 1.2 mL CLM24 cultures, incubated for 1 h at 37 °C and 220 rpm, and then washed twice with 0.85% w/v sodium chloride. The washed cell cultures were serially diluted, plated on LB agar, and incubated overnight at 30 °C before counting colonies.

TRI2-2 binding to the SARS-CoV-2 spike RBD protein was determined using AlphaLISA, based on previous descriptions[25]. AlphaLISA reactions were performed in a buffer containing 50 mM HEPES (Sigma-Aldrich H3375), 150 mM NaCl, 0.015% v/v TritonX-100 (Sigma-Aldrich 93443), and 1 g/L BSA (Sigma-Aldrich A2153) at pH 7.4. 2-μL reactions were prepared using a Beckman Colter Echo 525 liquid handler (Plate Reformat version 2.7.2), transferring solutions from an Echo Qualified 384-well PP PLUS plate (LabCyte PPL-0200) to a ProxiPlate 384-shallow well Plus plate (Revvity 6008280) using the 384_PP_Plus_AQ_GPSA fluid type. Serial dilutions of the His-tagged SARS-CoV-2 Spike RBD Protein (Acro Biosystems SPD-C82E9) and cell-free reactions incubated with either water or the TRI2-2_2xStrep plasmid were prepared using the AlphaLISA buffer. These dilutions were incubated with a final concentration of 0.02 mg/mL Anti-6xHis AlphaLISA acceptor beads (Revvity AL178C) for 1 h at room temperature before addition of Strep-Tactin AlphaLISA donor beads (Revvity AS106D) to a final concentration of 0.08 mg/mL. Following a second 1-h incubation, reaction luminescence was determined with a Synergy Neo2 plate reader with 80-ms excitation, 120-ms delay, and 160-ms integration time. Reactions were allowed to incubate for 10 min inside the plate reader before luminescence was recorded.

Trastuzumab binding was determined using a Trastuzumab Pharmacokinetic ELISA Kit (GenScript L00970). Cell-free reactions containing expressed trastuzumab were diluted 1:3,333 in PBS and assayed according to kit instructions.

To determine luciferase activity, cell-free reactions with expressed luciferase were diluted tenfold and mixed with ONE-Glo Luciferase Assay System (Promega E6120) in a 15:1 ONE-Glo to diluted reaction ratio. Luminescence was measured on a BioTek Neo2 plate reader for 30 min at 26 °C. The maximum recorded luminescence during this time period was recorded and used as a proxy for total active luciferase content in the reaction.

## Metabolite analysis

Cell-free gene expression reactions were run to monitor changes in key metabolites over the course of the reaction. At each time point, a set of reactions in triplicate was quenched 1:1 with 10% trichloroacetic acid and flash frozen in liquid nitrogen. Samples were then thawed and centrifuged at 20,000 $g$ for 10 min. to remove precipitated protein. The supernatant was collected, and 5 μL was injected onto an Agilent 1260 HPLC system (ChemStation version B.04.03). Metabolites were separated with an Aminex HPX-87H organic acids column at 60 °C with an isocratic flow of 5 mM sulfuric acid (Fisher A300-500) at 0.6 mL/min. Metabolite concentration was determined via refractive index detector based on the retention time and intensity of each compound's standard solution.

ATP was measured using the Promega CellTiter-Glo 2.0 Cell Viability Assay (G9241). Briefly, the quenched cell-free reaction supernatant was diluted 1:500 in nuclease-free water and mixed 1:1 with the CellTiter-Glo reagent; the luminescence was read on a BioTek Synergy H1 plate reader (Gen5 version 3.14). ATP concentrations were determined based on a standard curve prepared with pure ATP. Inorganic phosphate concentrations were determined with the Sigma-Aldrich Phosphate Assay Kit (MAK308) using a 1:500 dilution of the quenched cell-free reaction supernatant.

Reaction pH was measured at the 15-μL scale using an Orion ROSS PerpHecT pH electrode. Cell-free reactions used for pH measurements were not quenched with trichloroacetic acid nor flash frozen in liquid nitrogen.

## Reagent optimization strategy

We conducted our reagent optimization campaign across two primary stages: Exploration and Optimization. This workflow is outlined in Figures S1–S2. Round-by-round details regarding selection of tested reagent formulations are provided in Supplementary Data File 1 and all tested formulations are listed in Supplementary Data File 2.

Design of Experiments was used to explore the reagent formulation combinatorial space during the Exploration stage. Definitive screening designs were generated by JMP Pro 16 using the ranges of factors described in Supplementary Data File 1. All factors were numeric and continuous, and no blocks were added to the design. The goal of each definitive screening design was to maximize the yield of sfGFP. Run order was randomized before wet-lab experimentation. In two cases, response surface designs were generated using either the Central Composite Design or Box-Behnken design type, as described in Supplementary Data File 1. All factors were treated as continuous, run order was randomized, and number of center points was set as default.

Machine-learning guided Experimental Trials for Improvement of Systems (METIS)[59] was also used to explore reagent formulations. METIS was implemented using the Google Colab Optimization Note-Book provided by Pandi et al.[59]; no custom code was written. Only the parameters detailing reaction volume, fixed reaction volumes, number of days, and concentration limits were edited; details on these parameters and the Day 0 data used to initialize the workflow are provided in Supplementary Data File 1. METIS was run for three rounds each time the tool was implemented.

Following wet-lab experimentation, collected data was used to fit a variety of models using the JMP Pro 16 interface. Model personalities, stopping rules, and data used to fit the models for each round of experiments are described in Supplementary Data File 1. Protein yield was assigned as the Y role variable unless otherwise noted. All fits were run in the forward direction for variable selection (two-stage forward for generalized Poisson regressions) with the Rules option set to Combine to avoid nonsignificant interaction terms. No variable

estimates were locked during the fitting process. After fitting the data, JMP was used to run the model and the Prediction Profiler module was directed to maximize desirability (i.e., predict maximum protein expression). These predicted optimal reagent formulations were then tested in further wet-lab experiments. In some listed cases (Supplementary Data File 1), feature importance as calculated by the JMP model was used to adjust component concentration ranges and inclusion.

**Statistics & reproducibility.** No statistical method was used to predetermine sample size. In most cases, a sample size of $n = 3$ was selected to enable standard deviation calculations; $n = 6$ was used for some bioreactor tests due to the higher observed error. No data were excluded from the analyses and the investigators were not blinded to allocation during experiments and outcome assessments. The experiments were not randomized. Unpaired two-tailed t-tests were used for all calculations of statistical significance. Error was displayed as +/- standard deviation from the mean of all data points. In cases where calculations were applied to the collected data, propagated error was calculated according to standard propagation of uncertainty formulas, with data means and standard deviations used as input. Data visualization was performed with GraphPad Prism 10.5.0 and Microsoft Excel version 16.43 or 2506.

### Reporting summary
Further information on research design is available in the Nature Portfolio Reporting Summary linked to this article.

## Data availability
Source data for all figures are provided with this paper in the Source Data file, including main text figures, supplementary figures, and unedited autoradiograms. Data regarding reagent optimization campaign design strategies and optimization formulas are provided in the Supplementary Information/Source Data files. The genomic sequencing data for the BL21 Star (DE3) Δgor strain has been deposited in the NCBI database under GenBank accession code JBTLPL010000001.1 (https://www.ncbi.nlm.nih.gov/nuccore/JBTLPL010000001). Source data are provided with this paper.

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

## Acknowledgements
This work was supported by the National Science Foundation (CBET -2341123) (R.A., M.C.J.); DARPA (W911NF-23-2-0039) (M.L.O., C.E.C., C.A.S., R.A., Z.M.S., G.R., J.R.S., A.S.K., M.C.J.); the Department of Energy (DE-SC0023278) (M.L.O., A.S.K., M.C.J.); and a grant of the Korea-US Collaborative Research Fund (KUCRF), funded by the Ministry of Science and ICT and Ministry of Health & Welfare, Republic of Korea (grant number: RS-2024-00468410) (R.A., M.C.J.). M.L.O. acknowledges support from the National Science Foundation Graduate Research Fellowship under grant no. DGE-2234667. ZMS acknowledges support from the National Science Foundation National Research Traineeship under grant no. 2021900.

## Author contributions
M.L.O., C.E.C., J.R.S., A.S.K., and M.C.J. contributed to the conceptualization of the study. M.L.O. completed the majority of the research. R.A. prepared the genetically engineered lysate source strain. C.A.S. performed the bioreactor experiments. Z.M.S. assisted with the AlphaLISA experiments. M.C.J., A.S.K., J.R.S., and G.R. supervised the research. M.L.O., A.S.K., and M.C.J. wrote the manuscript. All authors commented on and edited the manuscript.

## Competing interests
MLO and MCJ have filed an invention disclosure based on the work presented. MCJ has a financial interest in SwiftScale Biologics, Gauntlet Bio, Pearl Bio, Inc., Synolo Therapeutics and Ridge Bio. MCJ's interests are reviewed and managed by Stanford University in accordance with their competing interest policies. The remaining authors declare no competing interests.
