## [Transparent Peer Review file · Nature Communications]

Design-driven optimization of low-cost reagent formulations for reproducible and high-yielding cell-free gene expression

Corresponding Author: Professor Michael Jewett

Version 0:

Reviewer comments:

Reviewer #1

(Remarks to the Author)

This represents a major undertaking to systematically evaluate how to most inexpensively generate lysate-based cell free reactions, which is an endeavor that many in the field will find quite useful. The comprehensive cost comparison calculations along with tables of sourcing allows for reproducibility. Demonstrating that this approach is feasible across various classes of enzymes is also useful to the field. Also indicating that high protein yields can be achieved with non-phosphorylated energy sources is also valuable.

A few (mostly minor) comments:

--Is it clear exactly why oxygenation is so critical for lysate-based systems? Since it is a lysate, we typically don't think of it undergoing aerobic respiration. What are key cellular processes that require dissolved oxygen?

--A minor thing, but D/L designators (S26) should use small caps font rather than regular font for stereochemical designation.

--Line 528 p.24 a manuscript on the bioreactor is said to be in preparation. This should just be changed to unpublished or a link to a preprint should be included.

--While inclusion of the protein sequences in the SI in Supplemental Table 4 is valuable, what would be most valuable for being able to reproduce this data would be to include the sequence of the synthetic DNA ordered. Different codon optimizations of different synthetic DNA can change results and this would be in the biggest spirit of synthetic biology reproducibility.

Reviewer #2

(Remarks to the Author)

This manuscript presents an interesting and valuable study that addresses an important problem with a novel experimental-computational optimization approach. The results are promising and the overall contribution is significant. However, the clarity and transparency of the methodological description are not sufficient in several places. In particular, the screening and iterative optimization process is described only qualitatively and lacks detailed explanation. These aspects are central to the reproducibility of the work, and should be clarified before the manuscript can be considered for publication.

Major:

1. The optimization process (Fig 1c and S4) lacks transparency and quantitative criteria; the authors should clarify why the procedure was halted at 524 experiments and provide explicit convergence metrics or iteration decision rules

2. The authors did not clearly explain how many conditions were retained at each iteration of the screening process, nor the basis for retention. It remains unclear whether the decision to keep or discard conditions was driven primarily by yield, by cost, or by a combined metric. The authors should explicitly describe the number of formulations advanced at each stage and the criteria used, to make the iterative design process transparent and reproducible.

3. While the study demonstrates substantial reagent cost reductions, the optimization effort itself is not quantified. For a fair assessment of efficiency, the authors should report the total optimization burden (e.g., cumulative experimental time, reagent consumption, and overall cost of the campaign). Furthermore, it remains unclear whether the optimized formulation is specific to *E. coli* lysates or it could be transferred to other CFE systems such as *B. subtilis*. A quick experiment and discussion of generalizability would broaden the impact and practical relevance of the work.

4. The entire optimization campaign appears to rely on sfGFP as the sole reporter. While convenient, this raises concerns about reporter-specific bias, since GFP is unusually stable and efficiently expressed in *E. coli* cell-free systems. Indeed, when tested with other proteins later, many did not exhibit the same level of improvement as sfGFP. The authors should explicitly acknowledge this limitation and, if possible, provide evidence from a more diverse reporter set or discuss how the optimized formulation generalizes across different classes of proteins.

5. The manuscript does not benchmark its optimization strategy against existing state-of-the-art approaches, such as active learning or machine learning–guided wet-lab optimization workflows that have been applied in cell-free systems (<https://www.nature.com/articles/s41467-025-58139-0>; <https://www.nature.com/articles/s41467-020-15798-5>). Without such a comparison, it is difficult to assess whether the proposed method is uniquely efficient or merely one possible route. The authors should at least provide a discussion of how their approach differs from, or complements, AI/ML-driven strategies, and under what conditions it would be preferable.

Minor:

1. I suggest reorganizing the sub-figures of Fig. 1 to align with the descriptions in Results. It is a bit weird to go through the figure in an a-b-d-c-e order.

2. Figure S4 illustrates performance trends across iterations, but the selection and retention criteria for conditions in each round are not sufficiently described. It would be great to highlight the points to be selected for next iteration in the plots.

Reviewer #3

(Remarks to the Author)

The manuscript by Olsen and colleagues addresses the optimization of cell-free protein synthesis for low-cost, high-yield production of proteins. Though cell-free reagent optimizations have been described over the last couple decades, the complexity of the reagents, and the cost per gram of protein have not significantly improved and have hindered adoption for commercial applications. These long-standing problems are addressed by the results presented here that describe a clearly superior reagent formulation that possesses more than order of magnitude improvements in cost and yield when compared to standard approaches. In most places, the manuscript clearly describes the results, the results are convincing (including valuable cross-laboratory validations), and the conclusions are nicely defended. The results will be of broad value to the large community of scientists that are advancing cell-free approaches and can enable long-discussed but not yet realized approaches to distributed manufacture of proteins. Further, simplifying the procedure, reducing associated costs, and demonstrating the general applicability for protein production will serve to further increase the growing use of cell-free approaches for applications beyond protein production.

The manuscript could be improved in a few locations. The strategy for reagent optimization is difficult to follow and is not further detailed in the Methods section. This section of the manuscript would benefit from a supplemental figure that outlines the optimization strategy and the decisions that led to the resulting formulation.

An interpretative analysis of the resultant formulation would benefit the manuscript. What insights can be gained that can further sustain or optimize protein synthesis? Understandably, the optimizations are performed using a variant of GFP, a standard benchmark, and the resulting formulation is further evaluated on a broad range of other proteins including difficult to synthesize therapeutic proteins. Demonstration of the modular nature of the reagent formulation is valuable and demonstrates practical implementation of the formulation. This demonstration also highlights the well-recognized problem of lower yield when preparing larger proteins with structural modifications. Some discussion as to the strategy for improving yield for such proteins would be useful, such as whether reagent optimization would need to be repeated.

Of a minor note, there is reference to work that is in preparation (Sundberg et al. line 528 and Lay line 839) that should be addressed with appropriate citations or adequate detail. Also, in Supplemental Figure 11, why are results for RFopt not shown? A simple clarification is needed.

Version 1:

Reviewer comments:

Reviewer #1

(Remarks to the Author)

It appears that all issues brought up in the original review have been addressed and clarified.

Reviewer #2

(Remarks to the Author)

The authors have conducted extensive discussions and additional experiments that adequately addressed my concerns.

Reviewer #3

(Remarks to the Author)

My prior criticisms of the manuscript are adequately addressed. I recommend it for publication.

Reviewer #1 (Remarks to the Author)

This represents a major undertaking to systematically evaluate how to most inexpensively generate lysate-based cell free reactions, which is an endeavor that many in the field will find quite useful. The comprehensive cost comparison calculations along with tables of sourcing allows for reproducibility. Demonstrating that this approach is feasible across various classes of enzymes is also useful to the field. Also indicating that high protein yields can be achieved with non-phosphorylated energy sources is also valuable.

We thank you for celebrating our work as a "major undertaking" that many will find quite useful.

Is it clear exactly why oxygenation is so critical for lysate-based systems? Since it is a lysate, we typically don't think of it undergoing aerobic respiration. What are key cellular processes that require dissolved oxygen?

Thank you for raising this point of clarification. Even though cell-free lysates might not typically be thought of as needing oxygen, it turns out that oxidative phosphorylation is a key driver for ATP regeneration. Inverted membrane vesicles form during cell lysis that contain all the necessary components of the electron transport chain, including ATP synthase. We previously demonstrated this here: <https://www.embopress.org/doi/full/10.1038/msb.2008.57>

In the revised manuscript, we add a sentence to better describe the dependence on oxygen and reference our previous publication:

"This finding [of oxygen sensitivity] supports previous demonstrations that oxidative phosphorylation is active in crude cell-free lysates and required for sufficient energy regeneration⁶⁹."

A minor thing, but D/L designators (S26) should use small caps font rather than regular font for stereochemical designation.

Thank you for catching this error. We have corrected it.

Line 528 p.24 a manuscript on the bioreactor is said to be in preparation. This should just be changed to unpublished or a link to a preprint should be included.

We have removed the reference to unpublished work.

While inclusion of the protein sequences in the SI in Supplemental Table 4 is valuable, what would be most valuable for being able to reproduce this data would be to include the sequence of the synthetic DNA ordered. Different codon optimizations of different synthetic DNA can change results and this would be in the biggest spirit of synthetic biology reproducibility.

Thank you for this suggestion. We agree that providing the DNA sequence will improve reproducibility of our work and have updated Supplemental Table 4 accordingly.

Reviewer #2 (Remarks to the Author)

This manuscript presents an interesting and valuable study that addresses an important problem with a novel experimental–computational optimization approach. The results are promising and the overall contribution is significant. However, the clarity and transparency of the methodological description are not sufficient in several places. In particular, the screening and iterative optimization process is described only qualitatively and lacks detailed explanation. These aspects are central to the reproducibility of the work, and should be clarified before the manuscript can be considered for publication.

We thank you for highlighting our promising and significant contribution to the field. We also agree that we could have done a better job clarifying some of the methods, which we have addressed in our revised manuscript and detailed below.

Major:

1. The optimization process (Fig 1c and S4) lacks transparency and quantitative criteria; the authors should clarify why the procedure was halted at 524 experiments and provide explicit convergence metrics or iteration decision rules.

We have extensively edited both our results section description of the optimization process and the methods section to now include a detailed section on the optimization strategy. Further details are providing in Supplemental Data File 1, detailing formulation design strategies and model use, and Supplemental Data File 4, listing all tested formulations and associated costs and protein yields. In addition, we have added Figure S2, which visually describes the exploration and optimization strategy as a flow chart with a description of number of experiments, rounds, and decision-making criteria. These new supplemental materials are referenced throughout the text.

As for why we stopped at 524 experiments, we chose to stop based on meeting our goal of “less than \$100 in reagents per gram of protein to more closely match cell-based production costs^{57,58},” (see Introduction, paragraph 4). With any optimization there is a question of what is good enough. Thus, we chose to halt the optimization campaign and proceed with validating and characterizing RF_{opt} . We now include a sentence in the manuscript explaining our reasoning.

Specifically, we state:

“We halted the optimization campaign at this point, having achieved our goal of producing protein for less than \$100/g protein.”

2. The authors did not clearly explain how many conditions were retained at each iteration of the screening process, nor the basis for retention. It remains unclear whether the decision to keep or discard conditions was driven primarily by yield, by cost, or by a combined metric. The

authors should explicitly describe the number of formulations advanced at each stage and the criteria used, to make the iterative design process transparent and reproducible.

We agree that we could have been clearer about how conditions were retained through the optimization process. Our optimization process was split into several phases, each with different design rules. In the first phase (illustrated by Figure 1b and Figure S1), we used a series of definitive screening designs to assess 11 reagent components common in established formulations. Concentration ranges used in each design were updated based on reagent solubility limitations and sfGFP production observed in each round. At this stage, conditions were not retained between iterations. Instead, each new set of data was appended to the existing data set and used to fit predictive models using JMP.

After identifying the minimal formulation, we moved to an exploration phase (Figure 1d, Figures S1-S5, and Supplemental Data File 1) focused on understanding the impact of different reagent components. Each round introduced different reagents or combinations of reagents; only the minimal system, used as the base formulation, was transferred between rounds. However, reagent ranges and/or inclusion were adjusted between rounds based on observed sfGFP production. To enrich our explored reagent combinations, we also regularly used definitive screening designs to survey how new components impacted the formulation, used the collected data to fit JMP models, and tested the predicted optimal reagent concentrations. Full details are provided in the new Supplemental Data File 1. This stage was terminated after we tested all components and broad combinations based on previous literature and our domain expertise.

We shifted to the final optimization phase by selecting two top-performing formulations: the formulation producing the most sfGFP and the formulation with the lowest reagent cost per gram sfGFP. We then conducted several rounds of targeted modifications based on learned design trends from the exploration phase. Specifically, each round used the best-performing formulations by yield from the prior round as the base for further optimization.

The following text was added to the **Results** section to reflect the above information and the Comment #1 by Reviewer 2:

*We varied these components while maintaining cell extract and plasmid DNA at fixed concentrations. We started by identifying upper and lower concentration bounds for each component at which we observed a measurable level of sfGFP synthesis. Using these concentration ranges, we then carried out a Definitive Screening Design (DSD) to define 16 reaction formulations using the 11 components. We tested these formulations in the laboratory and fit a stepwise model with the resulting data. We next adjusted formulation ranges based on components the model found significant. We repeated the process across 7 experiments to test a total of 67 formulations identified by Design of Experiments methods, 5 formulations predicted to be optimal by the stepwise models, and an additional 75 formulations designed based on reagent solubility and observed behaviors (Stage 1a; Figs. S1-S2; Supplemental Data File 1). After 7 rounds and 147 total formulations (**Fig. 1b**), we identified an active reagent formulation (selected based on activity and minimized number of*

components) comprised only of potassium glutamate, nucleotide monophosphates, and amino acids.

After setting this initial constraint, all further optimization was evaluated based on formulation productivity except where noted. Inspired by previous multi-dimensional reagent optimization efforts⁶⁸⁻⁷⁰, we tested new reagent formulations (based on the minimal formulation) in parallel and independent experimental rounds defined by (i) Design of Experiments as described in our initial exploration, (ii) the METIS active learning platform⁵⁹, and (iii) our domain expertise to identify reagents to add or remove (Stage 1b; Fig. S2; Supplemental Data File 1). Cycling through 26 rounds, we tested 176 formulations identified by Design of Experiments, 116 formulations predicted by JMP models, 131 formulations predicted by METIS, and 338 formulations designed using our domain expertise.

The best-performing formulations based on protein yield and cost per gram protein were then used as the baseline for further rounds of targeted optimization using learned design rules, such as yield improvement through addition of ribose and HEPES. We tested 323 formulations across 10 rounds (**Fig. 1c**; Stage 2; Fig. S2; Supplemental Data File 1). Each round of optimization used the best-performing formulations by protein yield from the prior round as the new base formulation.

The following text was added to the **Methods** section to reflect the above information and the Comment #1 by Reviewer 2:

*We conducted our reagent optimization campaign across two primary stages: Exploration and Optimization. This workflow is outlined in **Figures S1-S2**. Round-by-round details regarding selection of tested reagent formulations are provided in **Supplemental Data File 1**.*

*Design of Experiments was used to explore the reagent formulation combinatorial space during the Exploration stage. Definitive screening designs were generated by JMP Pro 16 using the ranges of factors described in **Supplemental Data File 1**. All factors were numeric and continuous, and no blocks were added to the design. The goal of each definitive screening design was to maximize the yield of sfGFP. Run order was randomized before wet-lab experimentation. In two cases, response surface designs were generated using either the Central Composite Design or Box-Behnken design type, as described in **Supplemental Data File 1**. All factors were treated as continuous, run order was randomized, and number of center points was set as default.*

*Machine-learning guided Experimental Trials for Improvement of Systems (METIS)⁵⁹ was also used to explore reagent formulations. METIS was implemented using the Google Colab Optimization NoteBook provided by Pandi et al⁵⁹; no custom code was written. Only the parameters detailing reaction volume, fixed reaction volumes, number of days, and concentration limits were edited; details on these parameters and the Day 0 data used to initialize the workflow are provided in **Supplemental Data File 1**. METIS was run for three rounds each time the tool was implemented.*

*Following wet-lab experimentation, collected data was used to fit a variety of models using the JMP Pro 16 interface. Model personalities, stopping rules, and data used to fit the models for each round of experiments are described in **Supplemental Data File 1**. Protein yield was assigned as the Y role variable unless otherwise noted. All fits were run in the forward direction for variable selection (two-stage forward for generalized Poisson regressions) with the Rules option set to Combine to avoid nonsignificant interaction terms. No variable estimates were locked during the fitting process. After fitting the data, JMP was used to run the model and the Prediction Profiler module was directed to maximize desirability (i.e., predict maximum protein expression). These predicted optimal reagent formulations were then tested in further wet-lab experiments. In some listed cases (**Supplemental Data File 1**), feature importance as calculated by the JMP model was used to adjust component concentration ranges and inclusion.*

3. While the study demonstrates substantial reagent cost reductions, the optimization effort itself is not quantified. For a fair assessment of efficiency, the authors should report the total optimization burden (e.g., cumulative experimental time, reagent consumption, and overall cost of the campaign). Furthermore, it remains unclear whether the optimized formulation is specific to *E. coli* lysates or it could be transferred to other CFE systems such as *B. subtilis*. A quick experiment and discussion of generalizability would broaden the impact and practical relevance of the work.

Thank you for suggesting that we quantify the optimization burden (e.g., time and cost), which was less than \$10 when not accounting for researcher time. We believe this more than justifies the long-term benefits that will be achieved from continued use of the system in years to come. We now include the following in our manuscript:

“The optimization campaign required approximately 175 hours of wet-lab work, 37 mL of cell-free reactions across the 1,231 formulations tested in triplicate, and only \$8.35 for all small molecule reagents.”

We share your interest in understanding if the learnings are transferable to other cell-free systems, such as those utilizing *B. subtilis* lysates. Neither we nor our collaborators have *B. subtilis* in the lab, so we cannot run such an experiment. However, we now discuss the generalizability of our formulation to other platforms. Specifically, we write:

“Although it is unclear whether RF_{opt} can be applied to other bacterial or eukaryotic lysates directly, recent work suggests that some level of transfer learning may be achievable^{73,74}.”

4. The entire optimization campaign appears to rely on sfGFP as the sole reporter. While convenient, this raises concerns about reporter-specific bias, since GFP is unusually stable and efficiently expressed in *E. coli* cell-free systems. Indeed, when tested with other proteins later, many did not exhibit the same level of improvement as sfGFP. The authors should explicitly acknowledge this limitation and, if possible, provide evidence from a more diverse reporter set or discuss how the optimized formulation generalizes across different classes of proteins.

We agree that our optimization campaign is limited by using only the stable, easily expressed reporter protein sfGFP. We now state this limitation. We write:

Lastly, we sought to address potential reporter bias that stems from using sfGFP as the reporter protein during the optimization campaign. sfGFP is highly stable and simple to express and fold⁷⁷. We therefore wanted to ensure that RF_{opt} could be used effectively to synthesize a variety of more complex, commercially relevant protein products.

While sfGFP is a rather simple protein, this is why in the initial manuscript we expressed an additional 16 real therapeutic proteins including antibodies, quantified by radioactivity. **All** proteins performed better than the historical system *when compared on a reagent cost per gram protein produced basis*. That said, we agree that an even more diverse reporter set of proteins would be beneficial. We therefore tested expression of an additional 7 diverse proteins using both the PANOx-SP and RF_{opt} formulations. These proteins include luciferase, 5 non-therapeutic proteins, and one *de novo* designed protein. Results from these tests are now included in Supplemental Figure 24 and included in the main text as follows:

“We further demonstrated expression for seven diverse, non-therapeutic proteins in RF_{opt}, with 4 of 7 proteins achieving statistically higher soluble expression levels in RF_{opt} (Fig. S24). Looking holistically, all proteins had lower \$/g_{protein} costs in the RF_{opt} formulation than the PANOx-SP formulation.”

In terms of our test protein sets, we observed that 11 of 23 had statistically higher soluble expression levels (g/L) with the RF_{opt} system, 10 of 23 were statistically the same, and 2 of 23 performed worse. However, **all proteins performed better with RF_{opt} when compared based on a reagent cost per gram protein basis**. This is now explicitly stated:

“Across all proteins for which soluble expression levels (g/L) were quantified, 11 of 23 had higher expression levels with the RF_{opt} system, 10 of 23 were statistically the same, and 2 of 23 performed worse. However, all 23 proteins performed better with RF_{opt} when compared on a reagent cost per gram protein produced basis. Variability in overall expression yield may be linked to DNA design, optimal pH values, or folding rates.”

5. The manuscript does not benchmark its optimization strategy against existing state-of-the-art approaches, such as active learning or machine learning–guided wet-lab optimization workflows that have been applied in cell-free systems (<https://www.nature.com/articles/s41467-025-58139-0>; <https://www.nature.com/articles/s41467-020-15798-5>). Without such a comparison, it is difficult to assess whether the proposed method is uniquely efficient or merely one possible route. The authors should at least provide a discussion of how their approach differs from, or complements, AI/ML-driven strategies, and under what conditions it would be preferable.

We agree that we could better compare our optimization strategy to existing machine learning-guided wet-lab workflows. We therefore tested the reagent formulations identified in the two suggested papers to systematically compare with our formulations. Our new RF_{opt} system is multiple orders of magnitude less expensive than these reagent formulations and produces 3-4 times more sfGFP in a batch reaction. We include this data in Figure 1a, Table 1, and write the following:

We also tested the best performing formulations identified by two machine learning-based optimization approaches^{63,65}.

We agree that discussion of how our optimization strategy compares to AI/ML-guided strategies and conditions for use would be beneficial. We now state:

RF_{opt} outperforms previously optimized reagent formulations for E. coli cell-free expression, including formulations identified with machine learning⁷⁰ or active learning-guided⁶⁸ optimization strategies for the same or similar E. coli lysates. We found that adding and removing reagent components across our exploration, rather than optimizing a fixed set of components, was key to achieving RF_{opt}. By blending human intelligence (domain knowledge) and computational reaction design, we were able to more effectively optimize reagent formulations. While we did not find a single computational design tool to be superior, relying solely on computational predictions and a single, defined reagent set is not sufficient for achieving high yields. A combination of computer-guided optimization followed by targeted additive exploration based on project goals may be more accessible to laboratories without access to domain knowledge.

Minor:

1. I suggest reorganizing the sub-figures of Fig. 1 to align with the descriptions in Results. It is a bit weird to go through the figure in an a-b-d-c-e order.

Thank you for your suggestion. We agree that it is best practice to order and label figure panels based on how they are referenced in the text. While arranging this figure's sub-panels was challenging, we have adjusted the layout of Fig. 1 to better align with the descriptions in **Results**.

2. Figure S4 illustrates performance trends across iterations, but the selection and retention criteria for conditions in each round are not sufficiently described. It would be great to highlight the points to be selected for next iteration in the plots.

Thank you for this suggestion. We agree that this would be a helpful addition and have updated the supplemental figures and methods as detailed above.

Reviewer #3

The manuscript by Olsen and colleagues addresses the optimization of cell-free protein synthesis for low-cost, high-yield production of proteins. Though cell-free reagent optimizations have been described over the last couple decades, the complexity of the reagents, and the cost per gram of protein have not significantly improved and have hindered adoption for commercial applications. These long-standing problems are addressed by the results presented here that describe a clearly superior reagent formulation that possesses more than order of magnitude improvements in cost and yield when compared to standard approaches. In most places, the manuscript clearly describes the results, the results are convincing (including valuable cross-laboratory validations), and the conclusions are nicely defended. The results will be of broad value to the large community of scientists that are advancing cell-free approaches and can enable long-discussed but not yet realized approaches to distributed manufacture of proteins. Further, simplifying the procedure, reducing associated costs, and demonstrating the general applicability for protein production will serve to further increase the growing use of cell-free approaches for applications beyond protein production.

Thank you for your detailed analysis of the work and recognition of the broad value of our work to the community and field.

The manuscript could be improved in a few locations. The strategy for reagent optimization is difficult to follow and is not further detailed in the Methods section. This section of the manuscript would benefit from a supplemental figure that outlines the optimization strategy and the decisions that led to the resulting formulation.

Thank you for suggesting we clarify our reagent optimization strategy. We have extensively edited both our results section description of the optimization process and the methods section to now include a detailed section on the optimization strategy. In addition, we have added Figure S2 and the Supplemental Data 1 file which now visually and textually describe the exploration and optimization strategy as a flow chart with a description of number of experiments, rounds, and decision-making criteria. This supplemental Figure and Tables are now referenced in the results and methods.

An interpretative analysis of the resultant formulation would benefit the manuscript. What insights can be gained that can further sustain or optimize protein synthesis? Understandably, the optimizations are performed using a variant of GFP, a standard benchmark, and the resulting formulation is further evaluated on a broad range of other proteins including difficult to synthesize therapeutic proteins. Demonstration of the modular nature of the reagent formulation is valuable and demonstrates practical implementation of the formulation. This demonstration also highlights the well-recognized problem of lower yield when preparing larger proteins with structural modifications. Some discussion as to the strategy for improving yield for such proteins would be useful, such as whether reagent optimization would need to be repeated.

We agree that further interpretative analysis of the RF_{opt} formulation would be beneficial. We now write the following to add to this discussion:

“We also demonstrate with the RF_{opt} formulation that neither inhibitory phosphate concentrations nor ATP depletion are responsible for termination of cell-free gene expression in our lysates, suggesting that optimizing the system for stability of other small molecules, RNA, or cellular machinery may be necessary to extend reaction duration.”

Thank you for suggesting that we add more discussion of how to improve yield for large, complex proteins. We now include the following:

“Additional formulation optimization, such as through the addition of chaperones^{89,90}, metabolic proteins^{67,91}, crowding agents^{92,93}, and vesicles or nanodiscs⁹⁴, may help to improve production of complex, large, or uncommon therapeutic proteins, or those with structural modifications.”

Of a minor note, there is reference to work that is in preparation (Sundberg et al. line 528 and Lay line 839) that should be addressed with appropriate citations or adequate detail.

Thank you for catching this error. We have updated these references.

Also, in Supplemental Figure 11, why are results for RF_{opt} not shown? A simple clarification is needed.

Thank you for this clarification question. In Supplemental Figure 11, we sought to demonstrate that lysate prepared from the BL21 Star (DE3) Δgor strain allowed us to maintain similar expression yields when using the PANOX-SP system. Data presented in Figure 4b shows the corresponding results for the RF_{opt} formulation. The following sentence was added to the caption of Supplemental Figure 11 to clarify this point:

Corresponding RF_{opt} data is shown in Figure 4b.